# Conceptualising Wellbeing for Australian Aboriginal LGBTQA+ Young People

**Shakara Liddelow-Hunt [1,\*], Ashleigh Lin [1], James Hector Laurent Hill [1], Kate Daglas [1], Braden Hill [2], Yael Perry [1], Mirella Wilson [3] and Bep Uink [4]**

1  Telethon Kids Institute, University of Western Australia, Crawley, WA 6009, Australia
2  Kurongkurl Katitjin Aboriginal Centre, Edith Cowan University, Joondalup, WA 6027, Australia
3  School of Psychology, Murdoch University, Murdoch, WA 6150, Australia
4  Kulbardi Aboriginal Centre, Murdoch University, Murdoch, WA 6150, Australia
\*  Correspondence: shakara.liddelow-hunt@telethonkids.org.au

**Abstract:** It is likely that young people who are both Aboriginal and Torres Strait Islander and LGBTQA+ would be at increased risk for poor mental health outcomes due to the layered impacts of discrimination they experience; however, there is very little empirical evidence focused on the mental health and wellbeing of Aboriginal and Torres Strait Islander LGBTQA+ young people. The current study represents a qualitative exploration of wellbeing among Aboriginal LGBTQA+ young people. This study consisted of semi-structured interviews and focus groups with Aboriginal LGBTQA+ young people aged 14–25 years old in the Perth metropolitan area of Western Australia. Thematic analysis identified seven major themes that were significant to participants' wellbeing: identity, family, community, visibility, services, stigma and navigating.

**Keywords:** LGBTQA+; Indigenous; Aboriginal and Torres Strait Islander; qualitative; wellbeing; mental health; intersectionality





## 1. Introduction

The developmental period of adolescence and young adulthood is the peak risk for the onset of mental disorders [1,2]. However, some populations of young people are at higher risk for poor mental health and wellbeing outcomes than others. Two such populations are Aboriginal and Torres Strait Islander young people [3] and lesbian, gay, bisexual, trans, queer/questioning and asexual (LGBTQA+) young people [4,5]. The elevated risk is largely attributable to the impacts of discrimination and systemic oppression of these marginalized identities [6,7]. It follows that young people who are both Aboriginal and Torres Strait Islander and LGBTQA+ would be at increased risk for poor mental health outcomes due to the layered impacts of discrimination they experience, and that their experiences would be qualitatively different to other young people because of their location at the intersection of two marginalised identities [8,9]. However, there is very little empirical evidence focussed on the mental health and wellbeing of Aboriginal and Torres Strait Islander LGBTQA+ young people. This is despite Aboriginal and Torres Strait Islander LGBTQA+ community advocates calling for work in this area for several years, pointing to experiences of marginalisation, rejection and bullying, feelings of isolation [10–12] and the inadequacy of services in addressing these issues [13]. Community advocates have named these issues as directly contributing to a high risk of suicide among Aboriginal and Torres Strait Islander LGBTQA+ people [14,15].

Advocates also highlight the heightened risk for poor mental health in youth and the need to "stand up for the queer young people" [16]. This has additionally been well articulated in previous academic literature, which highlights the need for an intersectional approach that will capture the interconnectedness of Aboriginal and Torres Strait Islander and LGBTQA+ identities, and the layered effects of trauma and discrimination [17]. Both

researchers and community advocates have highlighted that the inadequacy of health systems that do not recognise diversity fail Aboriginal and Torres Strait Islander LGBTQA+ young people, leaving a gap in possible supports that could bolster their wellbeing [9].

It is misguided to commence building an evidence base on factors and potential interventions that could bolster Aboriginal and Torres Strait Islander LGBTQA+ youths' wellbeing without first coming to an understanding of what constitutes wellbeing for this group of young people. The concept of wellbeing is a somewhat ambiguous one, whose definition tends to differ depending on the context in which it is being used. For example, wellbeing has emerged in the Aboriginal and Torres Strait Islander health context as a way of thinking beyond deficit models of health that were imposed during colonisation, towards a holistic understanding that better aligns with Aboriginal and Torres Strait Islander ontologies [18]. In LGBTQA+ research, wellbeing is often engaged as a multi-dimensional and non-pathologising approach that can better encompass positive outcomes. For the purposes of this study, this ambiguity allows the authors to explore potential wellbeing and mental health factors among a population at the intersection of multiple contexts. By approaching this topic through an intersectional lens, we can see how the 'mainstream' (non-Indigenous) perspective most often employed in LGBTQA+ research misses important parts of the lived realities of Aboriginal and Torres Strait Islander community members and imposes a colonial view of what might be considered 'health'. On the other hand, 'Aboriginal' models of wellbeing without explicit consultation with LGBTQA+ people (e.g., Gee et al. [18]) similarly omit the experiences of Aboriginal LGBTQA+ people, who are rendered invisible in deference to the interests of 'typical blakfellas' (i.e., cisgender and heterosexual; [19]) ('Blak' and 'blakfella' are self-descriptors commonly used by the Aboriginal and Torres Strait Islander community). Additionally, in both communities, there is a tendency to preference the voices of older people, and youth perspectives are often seen as less legitimate or not representative of the rest of the community. As such, Aboriginal and Torres Strait Islander LGBTQA+ young people need to be able to define wellbeing on their own intersectional terms.

While prior research has not explicitly set out to understand what constitutes wellbeing from an Aboriginal and Torres Strait Islander LGBTQA+ youth perspective, research in adults points to some topics of importance and charts the matrix of systems of oppression, sources of strength and strategies of resistance that Aboriginal and Torres Strait Islander LGBTQA+ adults enact. Research in this area confirms that Aboriginal and Torres Strait Islander LGBTQA+ people experience high levels of discrimination [19,20], including homophobia and transphobia that is distinctly gendered and racialized [21]. We might contextualise this discrimination by understanding it as the enforcement of the colonial project of gender, in which supposedly 'natural' concepts of gender, sexuality and family are revealed to be colonial categorisations that continually reinforce the inferiority of Indigenous peoples [22,23]. Aboriginal and Torres Strait Islander people are not passive recipients of oppression, however, and enact resistance in multiple ways, including notably in digital spaces [24,25].

The only other empirical study to date reporting on the experiences of Aboriginal and Torres Strait Islander LGBTQA+ youth, including those under 18 years, is the Dalarinji project [26,27]. Despite the input from Aboriginal and Torres Strait Islander LGBTQA+ people, this work preferences the voices of older non-Indigenous people in interpreting Aboriginal and Torres Strait Islander LGBTQA+ youth's experiences, resulting in an analysis that, to paraphrase Collins [28], reflects a basic lack of familiarity with queer blak realities. By approaching this discussion through the lens of Indigenous Standpoint Theory, we understand that our work is intimately tied to the authors' standpoints [29]. Much in the way that Aboriginal people would introduce themselves in community, placing themselves among kin and country and defining the relationships and obligations they have to each other, the research team can better understand their role as researchers by identifying the positionality that produces their subjective worldviews and exploring their obligations to the Aboriginal and Torres Strait Islander LGBTQA+ community. The current study was

carried out by a research team consisting of a young queer Wajarri (first author), a straight cisgender Noongar woman, a gay cisgender Nyungar (Wardandi) man, a queer cisgender non-Indigenous woman and a straight cisgender non-Indigenous woman. Additionally, this paper has been co-authored by a queer transgender Ngarrindjeri man and a bisexual Gunditjmara person.

Finally, it is necessary to understand the mental health and social emotional wellbeing of Aboriginal and Torres Strait Islander LGBTQA+ young people within the context of Australian settler colonialism. The colonisation of Australia was devasting for Aboriginal and Torres Strait Islander peoples. A large proportion of the Aboriginal and Torres Strait Islander population were killed during colonisation, through frontier conflict, which included the massacre of women and children, as well as the accidental and deliberate introduction of diseases such as smallpox, measles and influenza [30]. Additionally, land and resources were stolen en masse and destroyed for economic purposes. These changes were enacted by settler colonists who saw their way of life as morally and intellectually superior, with little recognition of Aboriginal and Torres Strait Islander peoples as human. The 20th century saw the punitive control and surveillance of Aboriginal and Torres Strait Islander people under the auspices of 'protectionism', which was enshrined into legislation through various state-based policies, the most infamous of which was the Western Australian Aborigines Act of 1905. Many Aboriginal and Torres Strait Islander peoples were forcibly removed from their homes and interred in reserves, and children were stolen from their families and placed in institutions, including missions. In these institutions, these 'Stolen Generations' children were forbidden from practicing their culture or speaking their language, and many experienced emotional, physical and sexual abuse. Because of this, Aboriginal people are in the process of healing from generation upon generation of trauma, abuse and disconnection [30], and this is something Aboriginal young people are very aware of. Many Aboriginal and Torres Strait Islander young people are able to speak extremely articulately about the impacts of historical oppression on present-day mental health and wellbeing, and this is critically important to understanding the results of the current study.

This study aimed to explore how Aboriginal and Torres Strait Islander LGBTQA+ young people defined their social emotional wellbeing, as well as identify factors that young people say are important to their mental health and wellbeing.

## 2. Materials and Methods

This study was part of a larger national research project (Walkern Katatdjin (Rainbow Knowledge)) aimed at understanding the mental health and social emotional wellbeing of Aboriginal and Torres Strait Islander LGBTQA+ young people in Australia. The findings from this study were initially published in a community report [31] and used to inform the development and analysis of a national survey of Aboriginal and Torres Strait Islander LGBTQA+ young people's mental health, social and emotional wellbeing and experiences using health services. The project is overseen by a Youth Advisory Group made up of Aboriginal LGBTQA+ young people aged 14–25 years, and a Governance Committee made up of Aboriginal and Torres Strait Islander LGBTQA+ adults with expertise in service provision, advocacy and/or community activism. Ethics approval for the current study was given by the Western Australian Aboriginal Health Ethics Committee.

### 2.1. Participants

Participants were *n* = 14 Aboriginal LGBTQA+ people aged 14–25 years living in the Perth metropolitan area. As such, we will only be referring to Aboriginal people in the results, given that there were no Torres Strait Islander participants. Most participants were over the age of 18 years (*n* = 11), with only three participants aged 14–18 years old. The mean age of participants was 20 years old. Most participants were cisgender women (*n* = 10), three were cisgender men and one was a non-binary person. Most participants identified as bisexual, pansexual or gay, and one participant identified as being on the

asexuality spectrum. Participants were from various communities across Western Australia, mostly urban and regional areas, although a one participant had spent time in remote communities. Some participants (*n* = 3) had lived in other Australian states. There was representation from diverse Aboriginal communities and cultures, which participants themselves pointed out in focus group discussions.

*2.2. Participant Recruitment*

Study recruitment posts were shared on Facebook and Instagram on Aboriginal and Torres Strait Islander, LGBTQA+ and Aboriginal and Torres Strait Islander LGBTQA+ community pages; organisations' pages; and in private groups. Study flyers were shared in local health services, youth centres, universities and TAFEs, as well as on public notice boards. Information about the study was also shared through community networks via word-of-mouth.

All participants provided written consent prior to participating. This study obtained exception from parent/guardian consent for participants who were minors, as seeking this consent is potentially detrimental or dangerous for young people whose family do not accept LGBTQA+ people and may act as a barrier to participation for young people who are not 'out' to their parents. Minors completed an in-person screening, in which a member of the research team discussed the study with them and asked a set of questions in order to determine the young person's capacity to provide informed consent.

*2.3. Data Collection*

Data collection occurred via one-on-one semi-structured interviews (*n* = 11) and two focus groups (*n* = 7). Interview facilitators worked from a question sheet that was developed in consultation with the study's Youth Advisory Group but allowed participants to redirect the conversation to discuss what they thought was important to their mental health and wellbeing and the wellbeing of Aboriginal and Torres Strait Islander LGBTQA+ young people generally. This allowed for rich and organic conversations that gave participants more agency in defining the research topic. Interview questions focused on who the person was, how they identified their sexuality and gender, how they experienced their Indigeneity, their relationships with family and community, what factors influenced their mental health and wellbeing and their experiences with health services. The guide included 5 core questions about participants' identity, experiences and wellbeing (e.g., "How would you describe your sexuality and gender identity?" "If we're talking about social and emotional wellbeing for Aboriginal LGBTQA+ young people, what do you think is important to discuss?") and 3 questions about service experiences, which are outside of the scope of this paper.

Focus groups consisted of 3–5 participants and two facilitators. Five out of the seven focus group participants had already completed a one-on-one interview, and one participant completed an interview after their focus group. As such, focus groups were less concerned with eliciting personal narratives than giving young people a chance to come together and collaboratively reach a shared understanding about what was important to their wellbeing. Therefore, the facilitators used a separate focus group guide that was developed based on the interviews to date; however, discussion was primarily driven by participants' brainstorm of important topics at the beginning of the session rather than by the facilitators working from the question guide. Participants first introduced themselves (i.e., their name, mob (The term 'mob' is used among Aboriginal and Torres Strait Islander peoples to refer to someone's language group or 'tribe', although it may also mean family or community), country, sexuality, gender and why they were interested in the study) and agreed on several topics of importance that they wanted to discuss together, before starting an in-depth conversation. These were supportive spaces in which young people argued, agreed and educated each other, and several participants stayed in contact after the focus groups. After both the interviews

and focus groups, many participants commented that it was nice to meet 'other people like them' and that it was the first time they had a space for these discussions.

Interviews and focus groups were audio recorded and sent to a professional service for transcription. Transcripts were checked by a member of the research team for accuracy, which was especially necessary to correctly capture Aboriginal language words (e.g., 'wedjala'—white person). Transcripts were then sent to the participants to confirm whether they were happy with their data and give them the chance to change or remove anything they had said which they were unhappy with.

Most interviews were facilitated by a young queer Aboriginal researcher. Two interviews were led by a straight cisgender Aboriginal researcher. Along with being consistent with an Indigenous research methodology, this allowed for a more peer-to-peer feeling and meant participants could throw around Aboriginal language words, reference culturally-specific experiences or talk about pop-culture memes without having to explain themselves. Focus groups were facilitated by a queer Aboriginal person and a straight cisgender Aboriginal woman. Data collection took place in late 2019 and early 2020.

*2.4. Data Analysis*

Data were analysed collaboratively by the research team using thematic analysis [32,33]. Working collaboratively allowed for the research team to moderate personal biases while still letting lived experience inform the analysis and develop a more nuanced shared understanding of the data. The data analysis team was made up of a young queer Wajarri person, a straight cisgender Noongar woman, a gay cisgender Nyungar (Wardandi) man, a queer cisgender non-Indigenous woman and a straight cisgender non-Indigenous woman.

Analysis was inductive and intersubjective, with a focus on allowing participant perspectives to come through with minimal disruption, whilst finding commonalities between participant narratives that could inform future wellbeing research and interventions. Researchers familiarised themselves with transcripts and then coded themes using NVivo. The research team then met to discuss and refine the codes and identify relationships between themes. The analysis went through multiple rounds of coding and discussion to refine codes, re-check the themes and subthemes against the data and build nuance. The analysis was then checked by the study's Youth Advisory Group, Governance Committee and at a Community Forum (comprised of Aboriginal and Torres Strait Islander LGBTQA+ community members and representatives from local services) to ensure that findings resonated with community members and were therefore more likely to be reliable. The sample size was sufficient to address the research question given the depth and richness of the data [34].

*2.5. Methodological Rationale*

The research method was stylistically informed by yarning methodology [35] and was positioned solidly within an Indigenous research paradigm but influenced by traditional Western research methods. Terszack describes yarning as "a process of making meaning, communicating and passing on history and knowledge . . . a special way of relating and connecting with the Nyoongah culture" [36] (p. 90). Information that emerges from a research yarning circle is inherently relational in that is developed from interactions between yarning circle participants and between participants and researchers, and it is informed by the country and cultural values and norms on which the yarning occurs [37]. Upon critical self-reflection, the authors concluded that that this study's methodology sat somewhere in between yarning and Western qualitative research.

## 3. Results

Our thematic analysis identified seven major themes: (1) identity; (2) family; (3) community; (4) visibility; (5) services; (6) stigma, shame and fear; and (7) navigating. Together, these themes provide a foundation for understanding Aboriginal LGBTQA+

wellbeing. The theme of services will not be discussed here, as it sits within a larger context that is outside of the scope of this manuscript.

The first five themes were those identified by participants themselves when asked directly: "what do you think is important to the wellbeing of Aboriginal and Torres Strait Islander LGBTQA+ young people?" and found to be salient across most, if not all, interviews and focus groups. Potential themes identified by participants but not found to hold true across the dataset were not included in the final analysis. The final two themes (stigma, shame and fear; navigating) were identified by the research team as being present across the interviews and focus groups but not directly named as potential themes by the participants themselves. They were, however, couched in the language used by the participants.

The historical context of colonisation sits as an undercurrent beneath all themes and was mentioned frequently by participants as impacting on their wellbeing in a myriad of ways. Additionally, they were very concerned with the presumed high suicide rate among Aboriginal LGBTQA+ young people. Participants assumed that the intersection of disadvantaged identities would result in increased risk and wondered how much this was an unnamed issue in their communities.

Finally, it is worth noting that Aboriginal LGBTQA+ young people still contend with the 'typical' stressors young people face. There are significant parts of their lives that do not have a clear relationship to their Aboriginal LGBTQA+ identities, and as such they are excluded from this analysis. However, the multifaceted nature of Aboriginal LGBTQA+ young people's experiences is important for providing context to the results below.

### 3.1. Identity

Participants identified holding strong and proud identity as a major part of their wellbeing. Identity is defined here as young people's sense of self and understanding of who they are, along with the way they present themselves to the rest of the world. Identity was not immutable and was discussed as part of a journey or narrative rather than as a fixed construct.

#### 3.1.1. How Participants Experienced Their Identities

Many participants stated that they were happy for the way they identify to change over time but worried about receiving negative comments from other people, or their previous identity being seen as somehow 'less real' or 'incorrect' rather than reflective of what was true for them at the time.

> "So I mean as far as labels go, they can change and I expect that for myself, if I was to fall in love with a woman or whatever the case may be..."

> "I once like was seeing a guy and I was actually embarrassed to tell people I was because I was like, "Oh, people are gonna start saying, 'you turned her straight'" and all of this sort of stuff . . . People just don't really understand that sexuality is fluid and you don't have to be identifying between one or the other."

Some participants felt that they experienced their Aboriginal and LGBTQA+ identities together and this could be a source of strength for them or could give rise to distress if/when they found these identities in conflict. Alternatively, others felt that these identities were largely held separate, and they were unable to find spaces in which they could be simultaneously queer and Aboriginal—although this was something that they aspired to. As one participant said:

> "I want to be able to—anyone to be able to coexist with all aspects of their identity and not be having to choose one or the other to identify with more."

Participants experienced their Aboriginal identity in different ways and staunchly insisted on recognising the diversity of Aboriginal peoples, but some common components of Aboriginal identity were family, community, culture, country and history,

and these connections could be disrupted by homophobic and transphobic family and community members.

*"I have gone back up to the mission and stuff like that. So seeing where the women's business happens and all that but I just wanna feel more a part of that, and not so—that's my country but I'm not familiar with my country, if you know what I mean."*

Many participants spoke about connecting/re-connecting to community and country, largely due to the impacts of the 'Stolen Generations'. This could be a difficult and nerve-racking process that was frequently reliant on their relationships with older relatives and Elders, who were sometimes homophobic or transphobic. As such, discrimination against LGBTQA+ people complicated (or arrested) the process of healing from the impacts of colonisation and could make maintaining the relationships to community and country, which are such a large part of Aboriginal identity, treacherous.

*" . . . especially because I rely so heavily on my family to connect me to culture, because they're the ones who, in a way, kind of control—I'm waiting on them right now to take me, so if I cut ties with them . . . That would create a barrier to me connecting to country and culture . . . "*

*"I struggled a lot feeling like I don't belong in my community because I'll never be fully accepted because I am queer."*

### 3.1.2. Building a Proud Aboriginal LGBTQA+ Identity

Encouragingly, most participants were very proud of their identities, often in defiance of other people's negative attitudes. However, pride was often not a default state of being, but the result of a long process of building up this part of their wellbeing. Similarly, participants had to come to terms with their Aboriginality, in particular, their place within their Aboriginal communities. This is particularly challenging given the political complexities that are often at the heart of community concern and discourse. Participants also had to come to terms with what it means to be an Aboriginal person within a settler colonial context, including having to contend with ongoing prevalent negative stereotypes and racism.

*"I feel like my Aboriginal identity, my queer identity are two things that I'm very proud of separately and that I really celebrate them together."*

*"...there's a lot of lateral violence, as I said. People think that they can gate-keep blackness and I experienced a lot of that while I was growing up."*

*"...there's heaps of native title **** going on at the moment. So there's a lot of drama and my aunty doesn't really want me to get caught up in that."* (Native title refers to legislation by which Aboriginal and Torres Strait Islander communities may gain (some) control over their traditional lands [38]; however, the process of obtaining native title is long, resource-intensive and is known to cause division within communities)

*"I struggled with my cultural identity for quite some time because it's just hard to identify as Aboriginal because you have to deal with so much—there's so much negative connotations with it."*

Negative comments and views about Aboriginal and/or LGBTQA+ people articulated by community members could undermine participants' sense of strength in themselves. This is not a direct one-way relationship—pride could be impacted by discrimination but could also be a buffer against its mental health effects.

### 3.1.3. The Need for Information to Enable Self-Discovery

Participants underwent a self-discovery process of realising that they were sexuality and gender diverse, and then acquiring the knowledge and language to articulate their specific identity. This was particularly true for participants who were pansexual, asexual or

genderfluid, given how little awareness there is among the general population about the existence of these sexualities and genders.

While this process of self-discovery was scary or distressing for a few participants, the majority found it uneventful. Participants did not want any kind of one-on-one support with this process, but they did wish that more current, reliable and easily accessible information had been available to understand what sexualities and gender identities were possible for them. Most participants became aware of different sexualities and gender identities through conversation with friends, internet forums and social media, or queer communities they were involved with.

> *"But then I found a group near where I was living . . . so you can go and talk to people similar to your experience and I was like, "Oh, there's all these other pronouns and genders and sexualities," I was like, "Okay, this makes sense now. I can actually identify the way I want to."*

Several participants described trying to find information online and giving up because they were encountering unhelpful and inaccurate information.

### 3.2. Family

Participants' relationships with family contributed substantially to wellbeing for participants, both positively and negatively. While discussion was generally centred around what participants described as 'close' family (parents, stepparents, siblings and grandparents, and sometimes aunts, uncles and cousins), who had the most impact on their wellbeing, extended family also played a small role.

Family could be an important source of emotional support and made participants feel safe and loved by expressing their acceptance or normalising LGBTQA+ identities. When family expressed negative attitudes towards LGBTQA+ people, whether directed at participants or others, this had a significant negative impact on participants' mental health and sense of wellbeing.

> *"I think family, number one, number one, absolutely . . . it's really important that family is there to support you, and it's just sad that some people don't understand what that person is going through and they want support from family. And with work and your health and that, that will fall into place once you have the support of family . . . at the end of the day, my family accepts me, I'm happy."*

### 3.2.1. Family's Attitudes to Sexuality/Gender

Most participants described themselves as 'lucky' that their close family was accepting of them. A few described homophobic behaviour by siblings and grandparents, and one participant spoke about their parents continually refusing to accept their LGBTQA+ identity.

While some family members said that they accepted participants' LGBTQA+ identity, this did not always translate into their actions, and family often said or did hurtful or inconsiderate things without realising their impact. This might include making offensive jokes, not talking about participants' same-sex relationships, using inappropriate language and deadnames or questioning participants' identity. Participants felt that they could tolerate behaviour and that it was worth tolerating for the sake of those relationships but wished that their family had the education needed to adjust their behaviour.

> *"And then they've also started being more inclusive in talking about future partners, so previously, its' always just "him and he", but now when they're asking, they say, "they or them . . . So they're trying, and I really appreciate it, and it's really great to see, but there are moments that, you know, because they don't have to think about this, they can say things or do things or insinuate things that are quite hurtful."*

Additionally, older family members might use the racialized hardships they had experienced to dismiss or invalidate participants' concerns, or participants felt they needed to accommodate their family's trauma.

*"I can't sit here and talk about how hard my life is being a gay male when my mum and dad are both blackfellas. They weren't able to go through high school, they were called [racial slur] and the rest of it. It's hard to be able to—what I'm trying to say is that their trauma and their experiences that they've experienced growing up isn't discounted, so I think when it comes to things like this, it's on their own terms and it's got to be on their own terms."*

Fear of rejection appeared in many, but not all, of the discussions and was generally regarded as a 'worst case scenario'.

*"—my parents and my aunties—just so supportive. They're—honestly, without them, I don't know where I would be literally. So I'm just thankful that my parents are really engaged in my life."*

*"But even after that, I still held back from telling my mum and my stepdad because of other reasons 'cos I honestly thought I was gonna get kicked out . . . "*

Many participants' relationships with specific family members changed after they 'came out', often in ways that were subtle but significant to participants, including relationships ending or becoming more distant, participants learning to tolerate hurtful behaviour from family or participants educating their family members about LGBTQA+ people and issues.

### 3.2.2. Other LGBTQA+ Family

Older LGBTQA+ family members paved the way for participants, provided education and support and were considered role models. Similarly, participants acted in this way for younger family members, and many said they were motivated to be involved in this study so that they could make things better for their Aboriginal LGBTQA+ family and other young people.

*"For me, I think it's helpful for me because my sister was queer. I think it would've been a lot different if my sister wasn't because she was the first person who shocked the family. So I knew that I had her who I could talk to her it about and that made me know that it was okay and I saw how my family reacted to that which made me a lot more comfortable with myself."*

When other family members 'came out' before them, participants generally found that their 'close' family was more educated and more accepting, as it gave family time to 'get used to the idea' and they had already started the process of learning about LGBTQA+ people.

### 3.2.3. Education for Family

Generally, participants wanted their sexuality and gender normalised within their family but the lack of education about LGBTQA+ people, issues and inclusive language was a major barrier.

*" . . . I just was going to agree that normalising it and treating your partner as they would treat your sibling's partner, like, in a heterosexual relationship, just normalising it."*

There were no culturally appropriate resources available to them, meaning that participants often had to step into the role of educator themselves; however, many did not feel informed enough to be teaching others.

### 3.2.4. Process of Family Becoming Accepting and Supportive

Parents, in particular, often had to go through their own process of becoming accepting and supportive over time. After their initial reactions to participants' 'coming out', which could range from immediate acceptance, indifference or denial, they then had to learn more about their child's LGBTQA+ identity and figure out how to talk about it.

*"I think you have to go through like a transition period where they get used to it, 'cos for some time, I feel like my family thought that it was a secret and didn't tell anyone."*

Participants recognised that this could be difficult for their parents due to their upbringing, religious beliefs, lack of education about LGBTQA+ people or fears for their child's wellbeing, but participants appreciated that their parents were trying.

*" ... 'cos it's the stereotype of parents not wanting their kids to be LGBT and stuff like that, but I think often a lot of the times the case is that because they care for them, they don't want them to have a harder life ... "*

### 3.2.5. Extended Family

Attitudes towards LGBTQA+ people within families were highly variable, and most participants had some extended family who were supportive and others who were not. For several, their sexuality had fractured their relationships with members of their extended family, or they were worried it would. Many felt that they had to hide their sexuality around extended family, and this made those relationships uncomfortable. This is significant given the importance of 'extended' family in Aboriginal culture [39,40].

*" ... ever since I've come out as gay, I've never spoken to mum's family ... So I could see that as soon as I come out, they've withdrawn."*

*"It was a bit weird still and I, for the longest time, didn't know when we saw our extended families, my cousins, my aunties and all that. I didn't know if they knew so I didn't know whether I should sorta go back into the closet a bit and not talk about it."*

### 3.2.6. The Role of Family in Aboriginal Culture

Given that family is a pillar of Aboriginal culture and identity, many participants felt reassured that they were less likely to be rejected by their family for being LGBTQA+ because of the cultural necessity of maintaining those relationships.

*"I think the family response to an Aboriginal person being gay is a lot better than what white people take it ... because of our connection to family that we can't lose, "just because our son is gay, it doesn't mean that they're gonna be [rejected] ... "*

However, where family members were willing to accept or ignore participants' LGBTQA+ identities but not adjust their behaviour, participants received the additional mental health burden of tolerating some discriminatory attitudes while setting boundaries, as severing relationships with family would be counter-cultural or could limit their access to country and culture.

*"And it's so hard, blackfellas as well, it's not like white people where it's like, "Oh my God, I'm having nothing to do with my family ever again. They're so dead to me," and it's like ... Not the way I'm brought up."*

### 3.3. Community

Community was another major contributor to mental health and wellbeing identified by participants. Participants primarily discussed the Aboriginal community, although they also mentioned other communities that they belonged to or wanted to belong to, such as the LGBTQA+ community. There is an overlap between the themes of community and family given the closeness of these concepts in Aboriginal culture, through the broader definition of family and the way in which the Aboriginal kinship system makes community members kin.

### 3.3.1. Aboriginal Community Attitudes towards LGBTQA+ People

Participants largely perceived the Aboriginal community as homophobic and transphobic; however, their experiences within community varied. A few participants had firsthand experiences of rejection, ridicule or abuse within community, or had heard stories about these experiences from other Aboriginal LGBTQA+ people.

*"Yeah, 'cos up there, they [an Elder] told me that I wasn't allowed back on the community because of it. Otherwise, they have the right to flog me . . . I haven't been back since, so I don't plan on going back, 'cos then I have to face so many people."*

Others felt that the topic was 'swept under the rug' and 'rarely spoken about', and this made them feel lonely, like there was no place for them in the community, or like they had to hide that part of themselves.

*" . . . when you grow up in a culture that is so heavily against LGBT mob, you feel very alone, and so I can understand why people would commit suicide and all that sort of thing because you can't be who you are."*

*"But also you just feel kind of uncomfortable not—like, having to hide a part of yourself. And I feel like I hide a part of myself when I'm talking to people in the community back at home."*

However, for several of the participants, their communities were very accepting of LGBTQA+ people.

*"No one's ever said any derogatory terms in terms of my community... Not like this. No one's ever done that."*

One participant reflected on how their community's attitudes and their own had changed over time:

*" . . . coming back home this time has been—even though it was under not great circumstances, it's been just really healing and I guess me being a queer person has just been so much—so normalised with my family. And there's even other queer Aboriginal people in town and it's crazy because I didn't know that there were any others before I left and—I don't know, just listening to the gossip about them like you would hear gossip about any other drama happening in town, and just making it feel so normalised has just been really cool."*

Overall, participants' experiences indicated that LGBTQA+ acceptance in Aboriginal communities varies across regions, communities, and families, in contrast to participants' largely negative perceptions.

### 3.3.2. Fear of Community Rejection

In response to the attitudes they perceived in their communities, the fear of being rejected for being LGBTQA+ was a concern for many participants and caused them intense distress. For many, this was intertwined with other worries that they were not fully accepted by their Aboriginal community because they were fair-skinned, disconnected from culture or did not grow up on country.

*"And then it's even within our own community, even other Aboriginal people, you can easily be told, "Oh, you white, claiming you black. You're just claiming because you want those benefits or you want that scholarship or—" it's such a f\*\*\*ing never-ending battle, to be honest. And then being gay just adds that level of pressure like, "You're a gay Aboriginal?"*

Feeling that they did not belong was a barrier to participants giving back to their communities and helping other Aboriginal LGBTQA+ young people, which many participants stated as one of their goals.

### 3.3.3. Traditional Views of Sexual and Gender Diversity

Some participants described a history of LGBTQA+ belonging that aligned with other lore (e.g., marrying people from the right skin group (skin groups are classifications under the Aboriginal kinship system that inform the relationships Aboriginal people have with each other, including romantic relationships), child-rearing, performing men's or women's roles), whereas others recalled Dreaming stories ('Dreaming' or 'Dreamtime' is fundamental to Aboriginal cosmology. Dreaming incorporates stories, songs, dance and art, but also

creation, country, the kinship system and the lore and law of Aboriginal communities) that forbid homosexuality and traditions that enforced gender conformity. As a way forward, participants generally developed their own interpretations of culture or found different ways to engage with cultural practices.

> *"I've been told of personal stories and Dreamtime stories where any sort of homosexuality isn't accepted in the Indigenous Culture and that sort of thing, but I also know that it's a different day and age and really, I think, one of the bigger and more important things about Indigenous Culture is family and love, and I think that works way more stronger as a whole than just not accepting."*

### 3.3.4. Learning How to Participate in Culture as an LGBTQA+ Person

Being LGBTQA+ sometimes changed how participants could participate in their cultures, and participants generally relied on other community members, family and Elders to help them navigate this. For example, participants were concerned with how avoidance rules applied to same-sex relationships or participating in men's and women's business as a non-binary person (in Aboriginal culture, many practices are separated into 'men's business' and 'women's business' and are restricted to men or women, respectively).

> *"And I think also, culturally, it's been a little bit conflicting for me because in my Culture traditionally, my mum wouldn't be able to meet my partner. They wouldn't be able to sit across from each other. They won't be able to speak each other's names. But that will be if my partner were a man. And my mum has met partners of mine. And at first, I didn't know how to go about that, 'cos my mum's very connected to Culture and I didn't wanna put her in uncomfortable position because it's kind of a grey area. But I think I've gotten over that a little bit now, 'cos I've introduced her to my serious partners and I think she likes having that relationship with them."*

### 3.3.5. The Role of Elders

Elders played a substantial role in helping participants feel strong in their culture and, therefore, feel well. If relationships with Elders were jeopardised by Elders holding homophobic or transphobic attitudes, this had a significant negative impact on participants' wellbeing. Similarly, accepting and inclusive-minded Elders played an important and unique role in supporting participants. From their own experiences, participants reported that some Elders accepted LGBTQA+ identities and supported their LGBTQA+ young people, but others did not.

> *" . . . when I was coming out to my family and stuff like that, some of them were not very supportive and nice, and then my oldies—once I ran away crying, my oldies came up and they were like "Don't even worry about them," because that just shows how non-cultured they are, because back in our time, it's not a problem then. It's not a problem now."*

Rejection from Elders was distressing, and participants were concerned about homophobic and transphobic Elders cutting them off from community and culture. Several participants felt frustrated that they could not question the Elders' negative attitudes because of the cultural hierarchy in place.

> *"—they go joke around and call him [homophobic slur] all the time and it's like a joke, but the way she [Elder] said it that one day was just so horrible. I was just shocked, but it's hard to put an Elder in their place. You can't."*

> *"I think that some Elders as well might not be open to people who are queer and that can be a barrier because obviously they're like cornerstones of our community and culture, they hold so much knowledge."*

The Elders who held anti-LGBTQA+ views were more likely to be regarded by participants as 'fake' Elders—old people who considered themselves Elders but were self-appointed and more focused on helping themselves than others.

3.3.6. Impact of Christianisation on Attitudes towards LGBTQA+ People

Many participants grew up in Christian families and had experienced religion being used to justify homophobia and transphobia. Some were cautious around highly religious community members, while noting that this was a problematic reading of religious texts.

*"So I've had 'religious' family members turn around and say I wouldn't be born queer if my family had gone to church more. But on the same token, my super really super unbelievably religious nan—like if the pope said, "Jump off this bridge," she would jump off this bridge, she's so religious—she's also just turned around and said, "If God didn't want you to be that way, he wouldn't have made you that way, and it's just my duty to love you as per His command . . . ""*

Additionally, many participants were resentful of the Christianisation that occurred as part of colonisation and the trauma their families experienced in missions. This made it easier to dismiss discriminatory statements made from a religious standpoint, acting as a buffer against internalising negative views about their LGBTQA+ identities.

*"My mum had to live down here in [Aboriginal children's home] and obviously they were told they had to give up their Culture and they had to take on the religion and it was really strict and a lot of homophobia is ingrained in religion and ingrained in those beliefs. I would be curious to know how our Culture would've evolved and the attitudes towards people who are part of the queer community without that influence from colonisation and religion."*

3.3.7. Need for an Aboriginal LGBTQA+ Community

Participants stated that being part of an Aboriginal LGBTQA+ community would make them feel prouder, less alone and give them a chance to talk about experiences that other people could not understand. However, there were very few openly LGBTQA+ people in their own communities, and no Aboriginal LGBTQA+ community existed for them in Perth.

*"I wouldn't want anyone else to feel this way. So why do I wish that I had someone who felt this with me?... I definitely feel like if I did know other queer Indigenous people that I wouldn't be so uptight about everything. I'd probably feel a bit more free . . . "*

Most participants had several Aboriginal LGBTQA+ friends and/or family members, who provided valuable support and visibility, but they did not view this as a community. Several participants were trying to create this community themselves, with limited success.

*" . . . it's really great to have a friend to chat to about this stuff . . . We get each other on a deeper level and I think it's at that stage that it's really impactful and powerful 'cos I know I can rely on this mob to help me with things . . . nothing beats therapy from your black LGBT friends . . . "*

3.3.8. The LGBTQA+ Community

Participants liked being able to talk about sexuality and gender with their LGBTQA+ friends, but they did not feel supported by the broader LGBTQA+ community, who they described primarily in terms of Pride events and nightclubs that lacked Aboriginal representation. The 'whiteness' of these spaces often made participants feel uncomfortable or misunderstood and many had experienced racism in LGBTQA+ spaces.

*"I wanna say that being part of the queer community, I'm really proud of my sexuality and I would say that it's really good to belong to this community, but the queer community is predominantly white, ok, and some of them are racists so it's not that great."*

*"But then I definitely—I feel quite left out when I go to Pride events and I don't see a lot of Indigenous people."*

## 3.4. Visibility

Participants' wellbeing was supported by seeing people like themselves in the media and their communities. Additionally, improved visibility impacted on other factors important to their wellbeing, namely identity, family and community. Interestingly, participants only discussed positive representation in the media and did not mention negative representations of Aboriginal LGBTQA+ people (which may be reflective of the dearth of representation).

### 3.4.1. Need for Aboriginal LGBTQA+ Visibility

Seeing other proud Aboriginal LGBTQA+ people, particularly in their own communities, helped participants realise that it was possible to be simultaneously Aboriginal and LGBTQA+ and feel strong in their identities and less isolated.

*"Had I been exposed more before, I probably would've identified as gay from a much younger age but I just didn't know it was a thing that you could do and be."*

*"I didn't actually mention how I started to accept myself. It was basically through YouTube and the internet and seeing people out there who are proud of their sexuality and totally okay with it, that helped me be okay with it."*

Additionally, they felt safer and more optimistic about their future when they saw other Aboriginal LGBTQA+ people live happy lives and being accepted by family, friends and community.

*" . . . our aunty is our family, and I think down the track that really helped me be confident in my family being accepting of me . . . "*

However, they noted the severe lack of Aboriginal LGBTQA+ representation, agreeing with each other on one or two notable exceptions (e.g., the 'Tiddas' skit in the television show Black Comedy).

*" . . . it was just skinny white women, and so it doesn't resonate with you because you're having your sexuality represented but you're not having your culture represented . . . "*

Additionally, most of the visible Aboriginal LGBTQA+ people in their own communities were gay men, and several participants expressed that they wanted to see the rest of the LGBTQA+ community represented.

### 3.4.2. Visibility as Education

Visibility was not only directly important to participants' own sense of wellbeing, but also played an important role in educating other people and thus reducing the amount of discrimination that Aboriginal and/or LGBTQA+ people experience.

*"I think having social media and having TV shows and stuff about the queer community is really cool 'cos it kind of normalises it, but also kind of helps educate people who might not know things otherwise. And instead of trying to force them to learn and whatever, they can just go and read stuff online or things that come through their newsfeed that kind of educates them without them knowing."*

For a few participants, openly LGBTQA+ figures in their communities or in the media had already helped them educate and connect with straight cisgender people and made 'mainstream' environments more welcoming for them. This helped to reduce the work participants did 'explaining themselves', reduce stigma around Aboriginal and LGBTQA+ identities and reassure their loved ones.

In general, participants felt that much more education was needed to reduce the stigma around LGBTQA+ identities and relationships, particularly in schools. This included the need for culturally appropriate educational resources accessible to a range of community members, across different locations and formats.

*"I think also when you're trying to educate families or Elders and the older communities, making education accessible in places where they would be, 'cos some people might not*

*have access to the internet . . . So I guess at clinics having fliers, you know how they always have those little flyers out about different things or they have ads on TV that have Aboriginal people in it, maybe having a same-sex couple in that, and then just having the information and education there for people to educate themselves"*

### 3.4.3. Improvements over Time

Participants enthusiastically reported improvements in representation over time, with more people (both Aboriginal and non-Indigenous) in their communities openly identifying as LGBTQA+, signalling a generally more inclusive environment.

*"So all the year sevens and stuff that are coming to the school are all so much more open and that's really nice to see honestly."*

In the absence of good Aboriginal LGBTQA+ representation in their communities, participants often assumed that role themselves. This meant being vocally proud, but also being ready to challenge people who expressed homophobic and transphobic opinions.

*"I think it's a bit different now, and in fact, I would want to be out in public spaces and in my community to provide that representation for younger people, you know?"*

### 3.5. Stigma, Fear and Shame

Participants' wellbeing and mental health was negatively impacted by feelings of fear, shame and stigma associated with their Aboriginal and LGBTQA+ identities, which they had to actively manage and challenge internally.

### 3.5.1. Stigma and Discrimination

Stigma against LGBTQA+ people was experienced by participants as violence, insults and a lack of acknowledgment of their identity. Racism was discussed more often in terms of disadvantage, disempowerment and the far-reaching effects of colonialism, although some participants also discussed receiving overtly racist insults. As Aboriginal and LGBTQA+ people, participants experienced the stress of being 'attacked on multiple fronts'.

*"It's inter-crossed for people who wanna be extra-racist or extra-derogatory. I've been called a gay [racial slur] before."*

*"And it's so much worse because you're facing intercommunity violence on two different levels, and that's the intersection that I think a lot of people forget about."*

### 3.5.2. Fear and Shame

Stigma and violence against Aboriginal and LGBTQA+ people resulted in fear and shame, which were recurrent motifs through the participants' stories. In particular, they were (or had been) afraid of people findings out that they are LGBTQA+, being rejected by loved ones if they 'come out', losing their connection to community and culture and confrontation in their everyday lives.

*"So I had to go through eight years, like eight-nine years, of not knowing and just being scared all the time, that maybe I've said something or done something, that was gonna clue people in and then I was gonna be f\*\*\*ing ostracised and bullied and s\*\*\*."*

Some participants described themselves or other Aboriginal LGBTQA+ people using drugs and alcohol to cope with these feelings.

*"—with the drugs, I guess it was my way of just blocking it or drowning it and I guess that was just a coping thing for me . . . "*

Acceptance from family, friends and community as well as visibility helped to alleviate these fears, but the continued prevalence of racist and anti-LGBTQA+ attitudes meant that a level of continued caution was not only guaranteed, but necessary to their safety.

*3.6. Navigating*

Participants had to navigate their environment in order to stay safe and reduce conflict. This required actively assessing how/if they could present their identities in different situations and policing their own behaviour and expression, which was often exhausting.

> *"And it's awkward because you see a guy that you've been with the night before with his girlfriend and then you're just off to the next aisle at the shop. So, it's all about—what's the word?—sort of like navigating, navigating."*

3.6.1. Navigating 'Coming Out(s)'

Participants challenged the narrative of 'coming out' as a one-time event, instead arguing that it is a continual everyday process. For them, this meant questioning whether they should 'come out' in any given situation, how and when to do it and why it was necessary. Instead, participants felt that every 'coming out' was a small 'leap of faith' and they needed to cultivate resilience to manage this.

> *"I'm fully out, I would say. But it's kind of a weird thing, right? 'Cos you don't—Are you supposed to tell everyone as soon as you meet them? But yeah, I guess I'm out. I just never 'came out'."*

> *"How do I know ... which one of those motherf\*\*\*ers is gonna go, "Yeah, I'd love to meet your girlfriend," or which of them is gonna be like, "You've got a f\*\*\*ing girlfriend? Gross." You don't really know and I think that's just where you have to develop a thick skin and have a support system."*

Many participants liked to use subtle or off-hand strategies to 'come out', which they felt helped to normalise LGBTQA+ identities and relationships.

> *"I was like, "Oh, yeah, I had my first kiss today. It was with this girl I really like." And my parents were like, "What?" I was like, "Yeah." 'Cos I was like, "Straight kids don't do this. So what's the point? I'm not gonna f\*\*\*ing do it. This is bulls\*\*\*.""*

Some fair-skinned participants also expressed that they often needed to 'come out as Aboriginal' and as such the daily stress of 'coming out' was doubled for them.

> *" ... I kind of relate it actually to being a light-skinned Indigenous woman because I constantly have to come out and say that I'm Indigenous. And I also have to then constantly come out and say that I'm queer ... "*

3.6.2. Coming out to Family

Despite the everyday nature of 'coming out', most participants felt that coming out to family and close friends was a particularly important moment for them. Some participants delayed coming out for several years after realising they were LGBTQA+ because they were afraid of rejection or conflict, and others did not think it was necessary for other people to know. Many participants tried to predict what their family's reaction would be based on their friends' reactions, or the way that their family spoke about other LGBTQA+ people.

> *"I think one of the scariest things about coming out is you don't know how your parents are gonna react ... But if there are other people in the family who have been queer and are open and proud, you can just use that to gauge how they're gonna react a little bit and it's really helpful."*

Several participants mentioned that they were concerned about how to come out to extended family, and that support from their close family would help them with this.

> *" ... it's hard to come out to people and it's an awkward conversation that you don't wanna have and having a family member do that for you alleviates that pressure and that takes away that job for you. So you want your family members to tell people so that you can just carry on about your life ... "*

### 3.6.3. Protecting Themselves Every Day

Participants had developed strategies for identifying potential allies and people who were likely to be homophobic, transphobic or racist. This often meant 'testing' people by starting conversations about equal rights prior to the participant disclosing that they are LGBTQA+, and participants had a list of behaviours or traits that they generally associated with positive or negative attitudes. For example, potential allies were people with welcoming body language who treated other Aboriginal and LGBTQA+ people with respect. Conversely, participants tried to avoid engaging with people who were religious, 'traditional', expressed bigoted views or 'gave them funny looks'.

Many participants discussed feeling guilty that they hid their sexuality or gender to avoid conflict. They were also ashamed of not challenging negative comments about LGBTQA+ or Aboriginal people.

> *"I'm cautious as to who I say things to, even though I know I shouldn't because I definitely feel guilty, 'cos I'm like—I shouldn't have to feel bad or suss someone before revealing these things to them."*

Participants used humour to diffuse conflict and learned to 'not care what people think' in order maintain a strong sense of self when experiencing hostility and harassment.

> *"If someone straight comes up to me and says, "Oh, you're gay," and I'd say, "Yeah, do you wanna—?"... I do find it quite effective. And sometimes they'll laugh with you about it, sometimes they'll just shut up straight then and there and I'll walk away with a smirk on my face."*

> *"I definitely didn't know how to look after myself at all, so when I was upset I'd just curl into a ball in my bed, and just be upset for ages because some random kid in the back of the class said something mean. . . . but nowadays I'm a little bit more "Whatever, they can say what they want, who cares"."*

### 3.7. Relationships between Themes

The above themes inform each other in complex ways, with the potential for both positive and negative impacts on wellbeing. Participants' sense of identity was strengthened by support from community and family as well as visibility. In addition to supporting identity, visibility also helped family and community to be supportive and could reduce the negative impacts of stigma, fear and shame on identity and wellbeing generally. When family and community were unsupportive and participants were struggling with the effects of stigma, fear and shame, pride in identity itself could act as a buffer to protect their wellbeing. Finally, participants needed to navigate their relationships with family and community, but supportive family could also act in the role of assistant navigator and facilitate their relationships with their communities.

## 4. Discussion

### 4.1. Social and Emotional Wellbeing for Aboriginal LGBTQA+ Young People

This study outlines some significant aspects of mental health and wellbeing for Aboriginal LGBTQA+ young people. These findings point to the importance of identity, family, community and visibility to Aboriginal LGBTQA+ young people's wellbeing; the negative impacts of stigma, fear and shame; and the navigating work necessary to protect their wellbeing.

It is worth considering how these findings compare with other conceptualisations of wellbeing, in order to fully understand the unique experiences of those young people at the intersection of Aboriginal and LGBTQA+ identities. Wellbeing for Aboriginal LGBTQA+ young people as compared to their heterosexual cisgender Aboriginal peers is particularly relevant, given the current moves towards inclusivity being taken by the Aboriginal Community-Controlled Health sector (e.g., [41]). Gee et al. [18] outline a model of social and emotional wellbeing from an Aboriginal perspective, which has since become influential in how both researchers and health practitioners approach Aboriginal

health. 'Wellbeing' as an Aboriginal and Torres Strait Islander concept differs from Western understandings of mental health [42]. Gee et al.'s model incorporates seven domains of wellbeing: (1) connection to body; (2) connection to mind and emotions; (3) connection to family and kinship; (4) connection to community; (5) connection to culture; (6) connection to country; and (7) connection to spirit, spirituality and ancestors. This model also recognises the ways in which these connections are impacted upon by social, historical and political determinants.

The themes of family and community identified in this study overlap with the domains of connection to family and kinship and connection to community. Other domains of Gee et al.'s model, namely connection to culture; connection to country; and connection to spirit, spirituality and ancestors, only appear in participants' narratives as mediated by relationships with family, community and Elders. The themes of stigma, fear and shame as well as navigating might be broadly understood as social determinants in Gee et al.'s model, which is constructed to be able to take into account the impacts of racism and colonisations on social and emotional wellbeing. As such, there is room within this model to examine the impact of cisheterosexism even if the model was, in its development, primarily concerned with the impacts of racism.

Visibility is an important theme identified in the current study that is not present in Gee et al.'s model. Visibility is, however, a common theme in other literature concerning Aboriginal and Torres Strait Islander LGBTQA+ peoples. For example, the importance of visibility to young people's identity echoes work by Hill et al. [19], O'Sullivan [24] and Farrell [25] that discusses how young people's ability to see and express multi-faceted Aboriginal and Torres Strait Islander LGBTQA+ identities in online spaces supports their wellbeing by offering a sense of belonging and a site for resistance. Our findings also highlight the added dimension of visibility within young people's offline communities, courtesy of other Aboriginal LGBTQA+ community members that are personally known to them. The absence of visibility in Gee's et al.'s model suggests that the model does not fully encapsulate Aboriginal LGBTQA+ young people's wellbeing. The similarities, however, provide an excellent starting point for expanding Aboriginal and Torres Strait Islander conceptualisations of wellbeing to include LGBTQA+ people, and highlight the ways in which Aboriginal LGBTQA+ young people are connected to their communities and culture.

*4.2. At the Intersection of Sexuality, Gender Diversity and Aboriginality in So-Called Australia*

Most striking about the findings are the ways in which Aboriginal LGBTQA+ young people's wellbeing is jeopardised by the intersecting forces of settler colonialism and cis-heteronormativity. For example, most participants expressed in some way their need to reconnect with family, country and community because of how those ties were severed due to protectionist policy and, in particular, Stolen Generations. The prevalence of this idea is consistent with reports indicating that approximately 46% of Aboriginal and Torres Strait Islander people in Western Australia are direct descendants of Stolen Generations people [43] and many others were impacted by the extreme lengths their families went to in order to not be stolen. The intergenerational trauma and disconnection that results from the violations experienced by their parents, grandparents and great-grandparents is known to have a detrimental effect on the health of current generations [43]. However, healing from the impacts of Stolen Generations is complicated for LGBTQA+ young people by prejudice against sexuality and gender diversity, which makes their relationships with family and community fragile. The irony of this is that, as the participants themselves articulated, much of the prejudice they experienced stemmed from the Christian religion that was introduced during colonisation and forced onto their ancestors in missions (again, often in the context of protectionist policies like forced removal or Stolen Generations). Christianisation played a large role in the restrictions on sexual diversity and gender non-conformity introduced to Aboriginal and Torres Strait Islander people as part of the colonial project [22].

Even when homophobic and transphobic community members argue that LGBTQA+ identities are not part of Aboriginal culture, cultures are continually changing and in flux. As one participant stated:

*"I've been told of personal stories and Dreamtime stories where any sort of homosexuality isn't accepted in the Indigenous culture and that sort of thing, but I also know that it's a different day and age and really, I think, one of the bigger and more important things about Indigenous culture is family and love, and I think that works way more stronger as a whole than just not accepting."*

This cultural adaptability is something that the participants advocated for, while also observing with frustration that many older community members are highly resistant to change. This resistance is not unreasonable given the ever-present threat of cultural destruction in the settler colony but may come at the expense of Aboriginal LGBTQA+ young people. This is not to imply that no restrictions on gender non-conformity or same-sex relationships existed in Aboriginal communities prior to invasion, but that current discrimination cannot be untangled from the far-reaching impacts of colonisation.

What this means is that young people's efforts to heal from colonial violence are complicated by anti-LGBTQA+ prejudice, which is in many ways a product of colonialism. This is made more difficult by the assumption made (and challenged) by participants that Aboriginal communities are not accepting of LGBTQA+ people, potentially stemming from racist notions that Aboriginal peoples are not progressive/progressed. By taking an intersectional lens, we might understand how these unique experiences detract from the wellbeing of Aboriginal LGBTQA+ young people.

However, this is not an entirely negative story. To develop strategies to improve their wellbeing, it is necessary to also understand the strengths that exist among Aboriginal LGBTQA+ young people and the communities that support them. Our participants were strong, articulate and driven to help their communities. They were overwhelmingly proud of their Aboriginal and LGBTQA+ identities, and participants were able to overcome feelings of shame that resulted from stigma through their own resilience and the support of their families and communities.

Participants were mostly doing well physically, emotionally and spiritually, in contrast to the largely negative narrative that appears in ethnographic studies with Aboriginal and Torres Strait Islander LGBTQA+ adults (e.g., [44]). This may be indicative of recruitment bias (as young people in crisis likely would not be interested in participating in research), an effect of doing interviews with an Aboriginal LGBTQA+ facilitator or simply mean that positive change has occurred over time and younger generations are more accepted and supported than older Aboriginal LGBTQA+ people.

The support participants received from family and community members comes through as a potential catalyst for change within the community. The majority of participants were accepted and supported by their families, often to their surprise, and the security they felt because of the cultural importance of family stands out as a significant strength unique to Aboriginal LGBTQA+ families. Similarly, Elders emerge as a unique part of Aboriginal community structures with huge potential to provide support to their LGBTQA+ young people, especially in their ability to facilitate connection to other aspects of wellbeing, such as culture and country.

Finally, it is worth emphasising the substantial impact Aboriginal LGBTQA+ people have for each other in providing support, visibility and inspiration, as well as participants' drive to become role models within their own communities when they are ready.

While recognising the many strengths that emerge in this research, it is also important to respect how concerned young people were with the risks for this population and the discrimination they personally experienced, and we should not undermine the complexity of their lived experiences. It is necessary to hold in mind that (a) risk for poor mental health in this population is a consequence of oppression and (b) this is not inevitable. By understanding both risk and protective factors, it is possible to create change that is grounded in the strengths of the community.

*4.3. Limitations*

While a sample size of 14 is reasonable in qualitative research, it means that the data are not adequately reflective of the diversity of Aboriginal and Torres Strait Islander LGBTQA+ communities and experiences. Notably, most participants were cisgender women, meaning that the perspectives of gender diverse young people and men were not adequately captured here. As none of the participants were Torres Strait Islander, that perspective is absent from this study.

As with all qualitative research, the analysis was influenced by researchers' personal perspectives and existing literature about the topic. Particular to this study, the research team was entirely cisgender and mostly women, meaning that there is a level of gender bias present in the analysis. Additionally, all Aboriginal researchers were either Noongar or Yamatji and as such represent less cultural diversity than the participants, who were from various Aboriginal cultures across Australia. The controls in place, such as review of the results by the project's Youth Advisory Group and Governance Committee, help to ensure sufficient rigour but cannot and should not erase all possible bias.

This study is geographically specific (all participants were currently living in the Perth metropolitan area), and while several of our participants had lived in regional communities, they were all, at the time of the interview, living in a major city with corresponding increased access to services, LGBTQA+ community organisations and internet. In many ways, the unique landscape of Perth will have influenced their experiences and opinions on wellbeing. As such, these findings do not represent Aboriginal LGBTQA+ young people outside of Perth, and it should not be assumed that these findings can accurately inform interventions in other places.

*4.4. Implications*

By exploring important aspects of wellbeing for Aboriginal LGBTQA+ young people, this study points to potential avenues for intervention at an interpersonal, community and services level. In particular, findings suggest that Aboriginal communities and Elders should actively and vocally provide support to their Aboriginal LGBTQA+ young people, and that the families of Aboriginal LGBTQA+ young people seek out further education on LGBTQA+ identities and issues. Aboriginal Community-Controlled Organisations may be particularly well placed to support communities and families in accessing education. The results of this study point to the positive impact that support from family, community and Elders can have on wellbeing, and this strength could be leveraged to support Aboriginal LGBTQA+ young people, who sit at a particularly vulnerable intersection of the forces of cis-heteronormativity and colonialism. The current findings also imply that service providers and clinicians need to be more aware of these intersecting vulnerabilities and the way that they impact on mental health and wellbeing outcomes.

Additionally, both Aboriginal communities and LGBTQA+ communities could do more to promote the visibility of people with both identities, backed by genuine inclusivity and awareness of diversity.

*4.5. Future Research*

Further research should be done with Aboriginal and Torres Strait Islander LGBTQA+ young people in different locations across Australia, particularly in regional and remote communities. Additional research should also be done to capture the experience of trans and gender-diverse Aboriginal and Torres Strait Islander young people. Participants were particularly interested in understanding the experiences of non-binary young people in participating in cultural practices, which are often separated into men's business and women's business. We would also suggest trialling interventions designed to improve the wellbeing of Aboriginal and Torres Strait Islander LGBTQA+ young people based on the strengths identified here, for example, the positive influence of supportive family and community members in partnership with community organisations.

This suggested research should contribute to a rigorous empirical database, which should also include large-scale quantitative data about Aboriginal and Torres Strait Islander LGBTQA+ young people's health and wellbeing. As the results of this study suggest, there are a wealth of possible aspects of wellbeing that merit investigation.

Most importantly, this research needs to be led by Aboriginal and Torres Strait Islander LGBTQA+ people and driven by lived experience and Indigenous research methodologies. Aboriginal and Torres Strait Islander people are chronically over-researched, and as Aboriginal and Torres Strait Islander people with an additional marginalised identity (i.e., LGBTQA+ identity), they are at risk of becoming objects of anthropological interest to white researchers who lack insight into Aboriginal and Torres Strait Islander LGBTQA+ lived worlds or obligations to work for the benefit of Aboriginal and Torres Strait Islander communities. Research into this area must be responsible, ethical and community-owned.

## 5. Conclusions

This paper forms part of the emerging literature concerning the experiences of Aboriginal and Torres Strait Islander LGBTQA+ peoples, while additionally capturing the perspectives of youth at this intersection. The findings point to the importance of identity; family; community; visibility; stigma, fear and shame; and navigating to Aboriginal LGBTQA+ young people's wellbeing. These themes encompass the risks that young people routinely navigate but also their many strengths, which should be utilised in providing urgently needed care.

**Author Contributions:** Conceptualization, all authors; Methodology, all authors; Formal Analysis, S.L.-H.; Writing—Original Draft Preparation, S.L.-H.; Writing—Review and Editing, A.L., J.H.L.H., K.D., B.H., Y.P., M.W. and B.U.; Supervision, B.U., Y.P. and A.L.; Project Administration, S.L.-H.; Funding Acquisition, B.U., A.L., Y.P. and B.H. All authors have read and agreed to the published version of the manuscript.

**Funding:** This research was funded by the National Health and Medical Research Council (NHMRC) Targeted Call into Research into Indigenous Social Emotional Wellbeing, grant APP1157377.

**Institutional Review Board Statement:** The study was conducted in accordance with the Declaration of Helsinki, and the protocol was approved by the Western Australia Aboriginal Health Ethics Committee (HREC910).

**Informed Consent Statement:** Informed consent was obtained from all subjects involved in the study.

**Data Availability Statement:** The data presented in this study are available upon approval by the project's data governance mechanism. Please contact the corresponding author for further information. The data are not publicly available in line with the Principles of Indigenous Data Sovereignty.

**Acknowledgments:** We would like to acknowledge our Governance Committee; Youth Advisory Group; and project partners Wungening Aboriginal Corporation, Yorgum Healing Services, SHQ and First Peoples Rainbow Mob WA for their support of this project.

**Conflicts of Interest:** The authors declare no conflict of interest.

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
