# Peer review of "Conceptualising Wellbeing for Australian Aboriginal LGBTQA+ Young People"

_2673-995X, doi:10.3390/youth3010005_

Round 1

Reviewer 1 Report

This is an excellent article.  It is well written, logical presented and the research is well executed.  The findings are informative and very important.  It is an important contribution to the literature, and the LGBTA field.  The discussion raises important issues and is comprehensive.

Minor points,, only for your consideration if you deem appropriate 

can you clarify if the same interview guide was used when conducting the one-on-one literature and the focus group and why you used these two data collection strategy?

Was there a finding(s) that surprised the research team?  And why?  

Author Response

Response to Reviewer 1 Comments

Reviewer feedback: This is an excellent article.  It is well written, logical presented and the research is well executed.  The findings are informative and very important.  It is an important contribution to the literature, and the LGBTA field.  The discussion raises important issues and is comprehensive.

Minor points, only for your consideration if you deem appropriate

Point 1: Can you clarify if the same interview guide was used when conducting the one-on-one literature and the focus group and why you used these two data collection strategy?

Response 1: Thank you for your thoughtful review and kind feedback. To clarify this point, we have expanded on the discussion as below. While the interviews put more focus on personal narratives, the focus groups emphasised collaboration and therefore required a different question guide in line with the different focus. If the reviewer would like the guides included as an appendix, we are happy to provide them.

“Five out of the seven focus group participants had already completed a one-on-one interview, and one participant completed an interview after their focus group. As such, focus groups were less concerned with eliciting personal narratives than giving young people a chance to come together and collaboratively reach a shared understanding about what was important to their wellbeing. Therefore the facilitators used a separate focus group guide that was developed based on the interviews to date, however discussion was primarily driven by participants’ brainstorm of important topics at the beginning of the session rather than by the facilitators working from the question guide.(lines 188-196)

Point 2: Was there a finding(s) that surprised the research team?  And why? 

Response 2: This is a wonderful question. While the findings provided clarity about young peoples’ experiences, they weren’t surprising to the research team for two main reasons. Firstly, the findings do fit within the existing literature about Aboriginal and LGBTQA+ peoples’ wellbeing – this study helped to provide a more detailed picture of that relationship and how these fields intersect. Secondly, given the lived experience of the research team, who are all active members of their communities, we had an approximate idea of what to expect. That said, the research team were pleased to find that most participants had overall positive experiences and reported good wellbeing. We have chosen not to explicitly address this point within the article but feel that it is implicit within the discussion.  

Reviewer 2 Report

I Recommended acceptance

this

article without any changes.

Author Response

Response to Reviewer 2 Comments

Reviewer feedback: I Recommended acceptance this article without any changes

Response: Thank you for your review.

Reviewer 3 Report

Dear authors,

I would like to congratulate you for the valuable research you have carried out. This study is of interest and has significant social and cultural value.

Furthermore, this study represents a change of perspective, by focusing on the positive aspects of human development (wellbeing and the promotion of mental health), instead of approaches that are more centred on psychopathological aspects.

As main strengths of the investigation I highlight:

The subject relevance and novelty.

The conceptualization of the research subject (focus on the positive aspects of development).

However, I suggest the following changes to improve the overall quality of the manuscript:

Title: the title does not correctly reflect the content of the article because, although in the introduction both groups are mentioned and analysed (Australian Aboriginal and Torres Strait Islander LGBTQA+), the empirical study does not include Torres Strait Islander participants, so for the sake of scientific rigor, Torres Strait Islander should be removed.

Introduction: a better definition and conceptualization of wellbeing (the authors mentioned the concept of wellbeing but did not provide a clear definition of this concept).

Methodology: although the methodology is well explained and gives a clear understanding of the procedures that were followed by the investigators, I consider that it should include the period during which data collection had taken place. This section should also include the interview guide (number and content of the questions).

Limitations: a paragraph should be dedicated to the discussion of researchers' perspective bias and how their personal perspectives can influence research results, since the research team share the same gender perspective as the studied group.

Author Response

Response to Reviewer 3 Comments

Review feedback: I would like to congratulate you for the valuable research you have carried out. This study is of interest and has significant social and cultural value.

Furthermore, this study represents a change of perspective, by focusing on the positive aspects of human development (wellbeing and the promotion of mental health), instead of approaches that are more centred on psychopathological aspects.

As main strengths of the investigation I highlight:

The subject relevance and novelty.

The conceptualization of the research subject (focus on the positive aspects of development).

However, I suggest the following changes to improve the overall quality of the manuscript:

Point 1: Title: the title does not correctly reflect the content of the article because, although in the introduction both groups are mentioned and analysed (Australian Aboriginal and Torres Strait Islander LGBTQA+), the empirical study does not include Torres Strait Islander participants, so for the sake of scientific rigor, Torres Strait Islander should be removed.

Response 1: Thank you for your kind feedback, we’re very proud of the focus on positive aspects of development. The reviewer raises an excellent point. We have amended the title of the article to ‘Conceptualising wellbeing for Australian Aboriginal LGBTQA+ young people’.

Point 2: Introduction: a better definition and conceptualization of wellbeing (the authors mentioned the concept of wellbeing but did not provide a clear definition of this concept).

Response 2: Thank you for raising this point. The term wellbeing is used here in a deliberately ambiguous way to reflect its usage in the Australian Aboriginal and Torres Strait Islander health context and for the purposes of exploring its conceptualisation in this article. We have added an approximate definition and explained why we chose to use the term in this way, as below. Given that this is still somewhat vague, we are open to further discussion.

“The concept of wellbeing is a somewhat ambiguous one, whose definition tends to differ depending on the context in which it is being used. For example, wellbeing has emerged in the Aboriginal and Torres Strait Islander health context as a way of thinking beyond deficit models of health that were imposed during colonisation, towards a holistic understanding that better aligns with Aboriginal and Torres Strait Islander ontologies [18]. In LGBTQA+ research, wellbeing is often engaged as a multi-dimensional and non-pathologising approach that can better encompass positive outcomes. For the purposes of this study, this ambiguity allows the authors to explore potential wellbeing and mental health factors among a population at the intersection of multiple contexts.” (lines 49-58)

Point 3: Methodology: although the methodology is well explained and gives a clear understanding of the procedures that were followed by the investigators, I consider that it should include the period during which data collection had taken place. This section should also include the interview guide (number and content of the questions).

Response 3: We have added the period in which data collection took place to and an example of the questions asked. If the reviewer would like the interview and focus group guides included as an appendix, we are happy to provide them.

“Data collection took place in late 2019 and early 2020.” (line 216)

“The guide included 5 core questions about participants’ identity, experiences and wellbeing (e.g. “How would you describe your sexuality and gender identity?” “If we’re talking about social and emotional wellbeing for Aboriginal LGBTQA+ young people, what do you think is important to discuss?”) and 3 questions about service experiences, which are outside of the scope of this paper.” (lines 183-187)

Point 4: Limitations: a paragraph should be dedicated to the discussion of researchers' perspective bias and how their personal perspectives can influence research results, since the research team share the same gender perspective as the studied group.

Response 4: We have added a paragraph detailing the researchers’ perspective bias. Thank you for pointing out the need for reflexivity around gender perspectives.

“As with all qualitative research, the analysis was influenced by researchers’ personal perspectives and existing literature about the topic. Particular to this study, the research team was entirely cisgender and mostly women, meaning that there is a level of gender bias present in the analysis. Additionally all Aboriginal researchers were either Noongar or Yamatji and as such represent less cultural diversity than the participants, who were from various Aboriginal cultures across Australia. The controls in place, such as review of the results by the project’s Youth Advisory Group and Governance Committee, help to ensure sufficient rigour but cannot and should not erase all possible bias.” (lines 1054-1061)